# Glucocorticoid-Induced Hypokalemic Periodic Paralysis after Short-Term Use of Tenofovir with Hypophosphatemia: A Case Report

**DOI:** 10.3390/medicina58010052

**Published:** 2021-12-30

**Authors:** Yujin Shin, Yonglee Kim, Kyong Young Kim, Jong Ha Baek, Soo Kyoung Kim, Jung Hwa Jung, Jong Ryeal Hahm, Min Young Kim, Jaehoon Jung, Hosu Kim

**Affiliations:** 1Department of Internal Medicine, Gyeongsang National University Changwon Hospital, Changwon 51472, Korea; yugene91@gnuh.co.kr (Y.S.); axlrose@gnuh.co.kr (Y.K.); sanel2@hanmail.net (K.Y.K.); jongha100@gmail.com (J.H.B.); jungwarmheart@naver.com (J.J.); 2School of Medicine, Gyeongsang National University, Jinju 52727, Korea; 9854008@naver.com (S.K.K.); jhjung@medimail.co.kr (J.H.J.); jrhahm@hanmail.net (J.R.H.); 3Institute of Health Sciences, Gyeongsang National University, Jinju 52727, Korea; 4Department of Internal Medicine, Gyeongsang National University Hospital, Jinju 52727, Korea; minyung000@hanmail.net

**Keywords:** hypokalemic periodic paralysis, hypokalemia, tenofovir, glucocorticoid

## Abstract

Hypokalemic periodic paralysis (HPP) is a neuromuscular disorder associated with muscular dysfunction caused by hypokalemia. There are various causes of HPPs and rarely, HPP appears to be relevant to tenofovir or glucocorticoid treatment. There have been several case reports of tenofovir-related nephrotoxicity or tenofovir-induced HPP. However, a case report of glucocorticoid-induced HPP in a patient using tenofovir temporarily has not been reported. Herein, we report a case of glucocorticoid-induced HPP with short-term use of tenofovir. A 28-year-old man visited the emergency room with decreased muscle power in all extremities (2/5 grade). In their past medical history, the patient was treated with tenofovir for two months for a hepatitis B virus infection. At the time of the visit, the drug had been discontinued for four months. The day before visiting the emergency room, betamethasone was administered at a local clinic for herpes on the lips. Laboratory tests showed hypokalemia, hypophosphatemia, and mild metabolic acidosis. However, urinalysis revealed no abnormal findings. Consequently, it can be postulated that this patient developed HPP by glucocorticoids after taking tenofovir temporarily. This is the first case report of glucocorticoid-induced HPP in a patient using tenofovir. Clinicians who prescribe tenofovir should be aware of HPP occurring when glucocorticoids are used.

## 1. Introduction

Hypokalemic periodic paralysis (HPP) is a rare neuromuscular disorder associated with muscular dysfunction caused by hypokalemia [1,2]. Disease severity and degree of paralysis vary with time and range. HPP may last from several hours to several days, and respiratory muscles may be involved in severe cases [1]. Most cases of periodic paralysis (PP) are hereditary, usually with an autosomal dominant inheritance pattern. Most cases of acquired HPP are thought to be associated with hyperthyroidism. HPP can be accompanied by heavy exercise, fasting, or a high-carbohydrate meal. In addition, glucocorticoid treatment could be a trigger of HPP [1,2].

In rare cases, HPP may occur with tenofovir treatment. Tenofovir is a nucleotide reverse transcriptase inhibitor used to treat human immunodeficiency virus (HIV) infection since 2001. Tenofovir-related nephrotoxicity has been reported in 1% to 2% of HIV-positive patients [3], which is associated with Fanconi syndrome (FS) and bone disease [4,5,6]. Venkatesan and Bhogal et al. reported HPP with FS in patients using tenofovir for the treatment of HIV infection [7,8]. However, there has been no case report of glucocorticoid-induced HPP after taking tenofovir temporarily. Herein, we report a case of glucocorticoid-induced HPP with short-term use of tenofovir.

## 2. Case Presentation

A 28-year-old man presented to our hospital emergency room complaining of acute weakness involving all four limbs. He denied a history of fever, alcohol consumption, herbal medication, diarrhea, or vomiting. The patient had no previous history of neurological or renal disorders. He was diagnosed with a hepatitis B virus infection six months prior and was prescribed tenofovir. He took tenofovir for two months and then stopped on his own. One day before the visit to the emergency room, he was injected with betamethasone at a local dermatology clinic because of simple herpes. That night, he visited the emergency room with weakness and numbness in both extremities.

The patient was apyretic, with a respiratory rate of 16 breaths/min. Blood pressure was within the normal range. On physical examination, the muscle power was 2/5 grade in all extremities. No other localized neurological abnormalities were observed. Laboratory test results were as follows: serum potassium, 2.2 mmol/L; sodium, 137.6 mmol/L; chloride, 105.7 mmol/L; calcium, 9.0 mg/dL; phosphorus, 1.1 mg/dL; albumin, 4.3 g/dl. The initial venous blood gas analysis results were as follows: pH, 7.31; pCO2, 47 mmHg; bicarbonate, 24 mmol/L; lactic acid, 1.9 mmol/L, presenting as mild metabolic acidosis with a normal anion gap. Alkaline phosphatase, aspartate aminotransferase, alanine aminotransferase, and creatinine kinase levels were slightly elevated to 139, 27, 55, and 197 U/L, respectively. Blood urea nitrogen was 12.7 mg/dL and creatinine levels were 0.77 mg/dL, which were within the normal range. Urinalysis showed no pyuria, hematuria, glycosuria, or proteinuria. The analysis of spot urine electrolytes revealed: sodium, 81.0 mmol/L; potassium, 193.3 mmol/L; chloride, 269.5 mmol/L; osmolarity, 921 mosm/kg; the calculated transtubular potassium gradient was 28.9%. Thyroid function test and renin aldosterone levels were within the normal reference range (thyroid stimulating hormone, 0.54 mIU/L; free T4, 0.87 ng/dL; free T3, 3.18 pg/mL; aldosterone level, 3.5 ng/dL; basal renin level, 0.50 ng/mL/h.).

The patient’s potassium and phosphorus levels were replaced with intravenous supplements, leading to a dramatic improvement in weakness of both arms and the right leg within four hours. After 12 h, the patient’s muscle power was fully recovered. Serum potassium and phosphorus levels were maintained within the normal range two days after cessation of electrolyte replacement (Table 1).

## 3. Discussion and Conclusions

HPP is caused by changes in the intercellular distribution of potassium or a substantial depletion of potassium due to renal or extrarenal loss of potassium. There are various causes of HPPs. Rarely, HPP may occur following tenofovir or glucocorticoid treatment [7,8,9,10,11].

Glucocorticoids have a pleiotropic effect, which used to treat a variety of conditions, such as allergic, autoimmune disease, or organ transplant rejection. The commonly used glucocorticoids are hydrocortisone, prednisolone, methylprednisolone, and dexamethasone. These glucocorticoids have good oral bioavailability and are eliminated by hepatic metabolism and renal excretion [12]. HPP triggered by glucocorticoids is rare. There are several mechanisms by which glucocorticoids induce HPP. Glucocorticoids act indirectly on Na-K-adenosine triphosphatase (Na-K-ATPase) to transport potassium into cells. In addition, glucocorticoids induce insulin resistance, leading to hyperglycemia and hyperinsulinemia. Glucocorticoid-induced hyperinsulinemia can shift potassium into the cell by increasing the pool of Na-K-ATPase [13]. It can also lead to renal excretion of the potassium by mineralocorticoid effect. Genetic predisposition involving the mutation of the transmembrane ion channels of skeletal muscle, including KCNJ18, CACNA1S, and SCN4A has been shown in some affected patients [14]. However, these mechanisms cannot fully explain HPP caused by glucocorticoids. The mechanism underlying dexamethasone-induced HPP remains obscure.

According to previous reports, HPP was induced by glucocorticoids used in systemic chemotherapy, Graves’ disease, and trigger point injection [9,10,11]. In a previous case report, Wessel et al. reported a case of HPP after use of dexamethasone in 1985. The patient experienced HPP four times after injection of a mixed formula of dexamethasone and lidocaine. In this case, HPP was the primary, autosomal dominant inherited form and possibly caused by a combined effect of dexamethasone with a consequent hyperglycemia and lidocaine with a change in the excitability of the muscle membrane [11]. In 2012, Lee et al. reported that HPP occurred after the use of dexamethasone during the treatment of thyrotoxicosis caused by Graves’ disease [15]. Therefore, it can be thought that HPP may occur if there are risk factors such as genetic mutation, lidocaine, or thyrotoxicosis due to several other complex mechanisms.

In this case report, it is postulated that the temporary use of tenofovir acted as a risk factor for glucocorticoid-induced HPP. Tenofovir disoproxil fumarate is an oral prodrug of tenofovir. Following absorption, tenofovir disoproxil fumarate is rapidly converted to tenofovir, which is a competitive inhibitor of HIV-1 reverse transcriptase. Tenofovir is excreted by the kidney, with 20–30% being actively transported into the proximal tubule [16]. Tenofovir may cause nephrotoxicity by inducing proximal renal tubular dysfunction. The mechanism of tenofovir-induced nephrotoxicity is related to the concentration of tenofovir. The organic anion transporter 1 in the proximal tubule actively transports 20–30% of tenofovir in the blood [17]. When the efflux of tenofovir is decreased by the multidrug resistance protein-2, the elevated tenofovir concentration inhibits mitochondrial function, leading to impaired proximal tubular cell function [18,19]. Tenofovir-induced proximal renal tubular dysfunction can lead to impaired renal reabsorption, eventually leading to FS, which manifests as dehydration, hypokalemia, hypophosphatemia, metabolic acidosis, and osteomalacia.

Tenofovir-induced nephrotoxicity has rarely been observed in HPP. According to previous reports, two patients developed HPP following long-term tenofovir treatment for HIV infection. The patients used tenofovir for two and seven years, respectively, which led to hypokalemia with FS [7,8].

In contrast, the patient in this case report used tenofovir for only two months and then discontinued for four months. In addition, there was no evidence of FS on urinalysis. Because the blood tests showed hypophosphatemia and mild metabolic acidosis without abnormal findings on urinalysis, although the level of urine phosphorus was not measured, it was hypothesized that the patient had recovered from renal tubular damage. Under these circumstances, the patient was injected with a glucocorticoid, which may have exacerbated the hypokalemia and induced HPP. Although tenofovir-induced nephrotoxicity manifesting as HPP is rare, tenofovir-related HPP triggered by glucocorticoids has yet to be reported in the literature.

In conclusion, this is the first case report of glucocorticoid-induced HPP in a patient on temporary tenofovir use. Although glucocorticoid-induced HPP is rare, clinicians who prescribe tenofovir should be aware of HPP occurring when glucocorticoids are used.

## Figures and Tables

**Table 1 medicina-58-00052-t001:** Laboratory data.

	At Admission	12 h After Admission	2 Days After Admission
Total protein (g/dL)	7.7		5.9
Albumin (g/dL)	4.3		3.7
ALP (U/L)	139		105
AST (U/L)	27		23
ALT (U/L)	55		44
LDH (U/L)	166		128
CPK (U/L)	197		
Glucose (mg/dL)	175		105
BUN (mg/dL)	12.7		9.9
Creatinine (mg/dL)	0.7		0.81
Na (mmol/L)	137.6	139.0	140.7
K (mmol/L)	2.2	5.0	4.1
Cl (mmol/L)	105.7	107.8	107.0
Ca (mg/dL)	9.0	8.8	8.5
P (mg/dL)	1.1	2.9	2.8
TSH (mIU/L)	0.54		
Free thyroxine (fT4) (ng/dL)	0.87		
ABG PH	7.31	7.34	
ABG HCO3 + (mmol/L)	24	22	
Urine protein	Negative		
Urine glucose	Negative		

Abbreviations: ALP, alkaline phosphatase; AST, aspartate aminotransferase; ALT, alanine aminotransferase; LDH, lactate dehydrogenase; CPK, creatine phosphokinase; BUN, blood urea nitrogen; TSH, thyroid stimulating hormone; ABG, arterial blood gas.

## Data Availability

No new data were created or analyzed in this study. Data sharing is not applicable to this article.

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
