# Peer review of "Glucocorticoid-Induced Hypokalemic Periodic Paralysis after Short-Term Use of Tenofovir with Hypophosphatemia: A Case Report"

_medicina, 2021, doi:10.3390/medicina58010052_

Round 1

Reviewer 1 Report

This is an interesting case, but I have some questions to be clarified.

In case presentation, please describe urine phosphorus concentration and evaluate whether hypophosphatemia is due to out-over or low-intake.

Please give a figure of clinical course of the case. As tenofovir was used several months ago from HPP, time-dependent changes of blood and urine test is important for presentation.

In discussion and conclusion, authors wrote that glucocorticoids act indirectly on Na-K-ATPase to transport potassium into cells. I think this is mainly due to mineral corticoid effect of glucocorticoid, but dexamethasone has almost no mineral corticoid effect, authors must consider the mechanisms of dexamethasone-induced HPP. Please describe whether there is a reported case of HPP induced by dexamethasone.

Considering that the patient showed apparent hypophosphatemia and authors could not deny the existence of FS, I suggest the title not using “without FS” but using “with hypophosphatemia”.

Reviewer 2 Report

The authors reported one case of glucocorticoid-induced HPP in a patient who used tenofovir without Fanconi syndrome, which is meaning for clinical usage of tenofovir and glucocorticoids.

  1. Line 81, In word “Aldosterone”, the “A” doesn’t need to be capital.
  2. Line 85, Since one case is reported, please change “patients’” to “patient’s”.
  3. Please add information of pharmacokinetics of tenofovir and glucocorticoids in the paper.

Round 2

Reviewer 1 Report

Authors answered my questions appropriately.